# Assessment of the Genetic Structure and Diversity of Soybean (*Glycine* *max* L.) Germplasm Using Diversity Array Technology and Single Nucleotide Polymorphism Markers

**DOI:** 10.3390/plants11010068

**Published:** 2021-12-26

**Authors:** Abdulwahab S. Shaibu, Hassan Ibrahim, Zainab L. Miko, Ibrahim B. Mohammed, Sanusi G. Mohammed, Hauwa L. Yusuf, Alpha Y. Kamara, Lucky O. Omoigui, Benjamin Karikari

**Affiliations:** 1Department of Agronomy, Bayero University Kano, Kano 700001, Nigeria; hassibonline1@gmail.com (H.I.); zlmiko.agr@buk.edu.ng (Z.L.M.); ibmohammed.agr@buk.edu.ng (I.B.M.); 2Centre for Dryland Agriculture, Bayero University Kano, Kano 700001, Nigeria; sanusi.gaya@buk.edu.ng; 3Department of Food Science and Technology, Bayero University Kano, Kano 700001, Nigeria; ladsufy83@gmail.com; 4International Institute of Tropical Agriculture, Ibadan 200211, Nigeria; a.kamara@cgiar.org (A.Y.K.); l.omoigui@cgiar.org (L.O.O.); 5Department of Crop Science, Faculty of Agriculture, Food and Consumer Sciences, University for Development Studies, P.O. Box TL 1882, Tamale 00233, Ghana; benkarikari1@gmail.com

**Keywords:** DArT-seq, genetic diversity, population structure, soybean, AMOVA, Africa

## Abstract

Knowledge of the genetic structure and diversity of germplasm collections is crucial for sustainable genetic improvement through hybridization programs and rapid adaptation to changing breeding objectives. The objective of this study was to determine the genetic diversity and population structure of 281 International Institute of Tropical Agriculture (IITA) soybean accessions using diversity array technology (DArT) and single nucleotide polymorphism (SNP) markers for the efficient utilization of these accessions. From the results, the SNP and DArT markers were well distributed across the 20 soybean chromosomes. The cluster and principal component analyses revealed the genetic diversity among the 281 accessions by grouping them into two stratifications, a grouping that was also evident from the population structure analysis, which divided the 281 accessions into two distinct groups. The analysis of molecular variance revealed that 97% and 98% of the genetic variances using SNP and DArT markers, respectively, were within the population. Genetic diversity indices such as Shannon’s diversity index, diversity and unbiased diversity revealed the diversity among the different populations of the soybean accessions. The SNP and DArT markers used provided similar information on the structure, diversity and polymorphism of the accessions, which indicates the applicability of the DArT marker in genetic diversity studies. Our study provides information about the genetic structure and diversity of the IITA soybean accessions that will allow for the efficient utilization of these accessions in soybean improvement programs, especially in Africa.

## 1. Introduction

Soybean (*Glycine max* L. Merrill) is an important legume crop, providing approximately 68% of the world’s supply of protein-rich meals and serving as an excellent source of vegetable oil and renewable fuel. The average worldwide yields of soybean increased from 2183 kg ha^−1^ in 1994 to 2769 kg ha^−1^ in 2019, with an annual increase of 26 kg ha^−1^ [1]. This shows that there is still considerable potential for increasing soybean yield. The use of new sources of genetic variation to develop improved varieties is enhanced when the selection of parental genotypes is based on both phenotypic and genetic diversity [2]. Therefore, determining the variation in the soybean germplasm is important for the effective selection of parental lines for hybridization and the development of improved varieties.

Furthermore, the development of improved crop varieties that are resilient to the changing climate is one of the strategies for overcoming the challenges of climate change and population growth [3]. To achieve this, the genetic diversity of crops needs to be exploited and sustained [4]. Similarly, determining the genetic diversity available in the germplasm is crucial for developing new improved cultivars with desirable attributes. However, in the developing world, the low genetic diversity in the cultivated gene pool has significantly hindered soybean improvement [5,6,7,8]. Previous studies that were conducted on the genetic diversity of soybean utilized accessions from China and the USA [9], the Korean peninsula [10] and a few accessions from Africa [11]. Therefore, it is important to assess the genetic diversity of soybean germplasms comprising accessions from Africa for achieving a sustainable improvement in soybean yield and quality, as this crop is predicted to dominate crop production across Africa soon, due to its potential [12]. 

The genetic diversity of crops can be assessed at phenotypic and genotypic levels or by using statistical methods to partition phenotypic or genetic descriptors into genetic or environmental components [13]. The diversity among genotypes can be estimated using morphological markers [14]; however, these markers are less effective than DNA markers because of their limited number, subjectiveness and susceptibility to environmental influences [14]. 

The use of DNA markers to evaluate diversity aids in the effective utilization of germplasm for crop productivity improvement and conservation. The availability of a reliable, robust and economical marker platform is important for breeding programs aimed at utilizing genetic diversity. Genetic markers such as single nucleotide polymorphism (SNP) markers are more reliable in estimating crop diversities [15]. Recently, the application of diversity array technology (DArT), which is an advanced genotyping-by-sequencing (GBS) platform [16], has resulted in a more efficient method of detecting high-density SNP information with a significantly reduced time and cost [17]. Thus, studies of the genetic diversity of several crop species have a relatively better coverage and fewer missing data using DArT compared to other GBS platforms [18,19]. 

Recently, soybean germplasms from Africa have been reported to have a narrow genetic base, based on phenotypic evidence [20]. However, less information is available at the molecular level [21]. A few studies have reported the genetic diversity of soybean accessions from Africa using simple sequence repeats (SSR) [11], but there is no available information about these accessions based on GBS-SNP markers. Therefore, the objective of this study was to examine the genetic diversity and population structure of 281 soybean accessions of the International Institute of Tropical Agriculture (IITA) using DArT-SNP markers in order to gain a better understanding of their potential utilization in soybean improvement programs in Africa.

## 2. Results

### 2.1. Marker Quality and Genome Characterization

A total of 19,505 SNPs and 16,116 DArT markers were generated for the 281 soybean accessions. After the first stage of filtering, which excluded unmapped markers, 18,270 SNPs and 14,051 DArT markers were obtained. The second stage filtering returned 7244 SNPs and 4443 DArT markers. The SNPs were distributed across the 20 soybean chromosomes (Appendix Aa), with Gm18 having the highest number of SNPs (533), while Gm12 had the lowest number (261). For the distribution of the DArT markers on the chromosomes, Gm16 had the highest number of markers (388) followed by Gm18 (387), while Gm12 had the lowest number (126) (Appendix Ab). The polymorphism information content (PIC) ranged from 0.10–0.50 for both SNP and DArT markers (Appendix Aa,b, respectively). Other quality parameters are presented in Appendix A. For both the SNP and DArT markers, their minor allele frequency (MAF) ranged from 0.05–0.50 and their reproducibility ranged from 0.95–1.00 (Appendix A).

### 2.2. Population Structure of the Soybean Accessions

A cluster analysis of the SNP and DArT markers revealed two groups, with each group having sub-groups (Figure 1A,B, respectively). For the SNP data, Group I had four sub-groups, with the USA accessions scattered throughout all the sub-groups, while Group II had 10 sub-groups (Figure 1A), comprising accessions from Taiwan and China. The DArT cluster analysis also revealed that Group I had nine sub-groups, and Group II had five sub-groups (Figure 1B). The accessions from the different countries were scattered throughout all the sub-groups. Principal component analysis (PCA) was conducted for the SNP and DArT markers (Figure 2A,B). The PCA for the SNP markers revealed two groups, with Group 2 having two sub-groups (Figure 2A). The accessions from the USA were scattered over both groups with the majority in Group 1, while the accessions from Africa were mainly in Group 2. Group 2 was further divided into two sub-groups, which were similar to Group 1 accessions. A similar trend was observed for the PCA of the DArT markers (Figure 2B). 

The model-based (Bayesian) analysis of the SNP markers estimated two groups (Q1 and Q2) (Figure 3A), corresponding to the best-fit K-value of 2 (Figure 3B). There were 89 accessions in Q1 and 192 accessions in Q2, with cluster values 0.17 and 0.83, respectively. Most of the accessions from Africa were in Q1, while other accessions were in Q2, indicating a clear differentiation between the origin of the African accessions and accessions from other countries. The mean fixation index (*Fst*) was 0.23 and 0.40, while the expected heterozygosity was 0.45 and 0.33 for groups Q1 and Q2, respectively, (Table 1) indicating a high level of genetic variation. The allele frequency divergence (net nucleotide distance) between Q1 and Q2 was 0.10. The accessions in Q2 are sub-divided into two sub-groups, similarly to the cluster and PC analyses.

The model-based analysis of the DArT markers also estimated two groups (Q1 and Q2) (Figure 3C), which was supported by the Evanno criterion best-fit K-value of 2 as the highest population structure (Figure 3D). There were 227 accessions in group Q1 and 54 accessions in Q2 at cluster values of 0.32 and 0.63, respectively. Most of the accessions from Africa were in Q2, while accessions from countries outside Africa were in Q1. The mean fixation index was 0.32 and 0.19, while the expected heterozygosity was 0.27 and 0.30 for groups Q1 and Q2, respectively (Table 1). The allele frequency divergence (net nucleotide distance) between Q1 and Q2 was 0.09. The accessions in Q1 were sub-divided into two sub-groups corresponding to the clustering and PC analyses.

### 2.3. Analysis of Molecular Variance

The analysis of molecular variance (AMOVA) based on SNP markers revealed that 97.00% of the genetic variance was within the population and 3.00% was among populations (Appendix Aa). The PhiPT value of the estimated population structures was 0.03 (*p* < 0.01), suggesting a high level of difference, and the number of migrants (Nm) was 18.02 (Table 2). The DArT markers revealed that 98.00% of the genetic variance was within the population and 2.00% was attributable to variance among populations (Appendix Ab). The PhiPT value was 0.025 (*p* < 0.01), while the Nm was 19.54 (Table 3).

### 2.4. Allelic Patterns across the Populations

Based on SNP markers, Shannon’s diversity index was used to quantify the genetic diversity of the 281 soybean accessions. The Shannon’s diversity index ranged from 0.13 in population 4 (Chile) to 0.52 in populations 5 and 13 (China and USA, respectively) (Appendix A). The lowest number of effective alleles was 1.19 in population 4 (Chile) and the highest was 1.59 in population 5 (China), followed by 1.58 in population 13 (USA). The highest number of alleles (Na) was 2.00 in population 13 followed by 1.98 in population 5. The diversity (h) was highest in population 5 (0.34) followed by population 13 (0.34), while the unbiased diversity was highest in population 5 (0.35).

Based on the DArT markers, Shannon’s diversity index ranged from 0.09 in population 11 (Thailand) to 0.64 in populations 10 and 13 (Taiwan, Province of China and USA, respectively), followed by 0.62 in population 5 (China) (Appendix A). The lowest number of effective alleles was 1.29 in population 9 (Senegal), and the highest was 1.71 in population 10 (Taiwan, Province of China) followed by 1.69 in populations 5 and 13 (China and USA, respectively). The Na was highest at 2.77 in populations 5 and 10. The value of h was highest in populations 5, 10 and 13 (0.37), while uh was highest in population 7 (0.42).

## 3. Discussion

Studies of genetic diversity of the IITA soybean accessions are important for conservation and utilization. These studies will assist soybean breeders in the development of suitable procedures in attaining wide diversity in soybean breeding programs in Africa [21]. Analyses of the genetic diversity and structure within a population of individuals are performed to determine the magnitude of genetic variations and their usage in breeding programs, e.g., marker-trait associations and linkage analyses. Recently, molecular markers such as the GBS-SNP markers have been used in estimating genetic diversity and have been found to be particularly efficient [15]. Notable among the GBS platforms is DArT, which generates large numbers of SNPs that are used in genotyping and genetic analyses [22]. Thus, in this study, the genetic diversity and population structure of soybean accessions from different countries were investigated using the DArT GBS technology. The DArT has the advantages of rapidity, low cost and high throughput and has been utilized to study the genetic diversity of some crops [23]. This technology has been applied in a recent study on the genetic diversity of rice, revealing a high genetic variation [24]. The current study is the first report on a successful application of DArT-seq for investigating the genetic diversity of soybean accessions, including those from Africa, using 7244 SNP and 4443 DArT-seq markers. The SNP and DArT markers were found to be effective in describing the genetic diversity of the 281 soybean accessions. Previous studies conducted on the genetic diversity and population structure of soybean used accessions from China and the USA [9], and the Korean peninsula [10].

In the present study, 19,505 SNPs and 16,116 DArT markers were analyzed and filtered to remove redundant markers that may not be functional. After filtering, 7244 polymorphic SNPs and 4443 DArT SNPs were retained and distributed across the 20 chromosomes of the soybean genome. Redundant markers are mostly not useful for specific genetic analyses, as reported previously [25,26]. Furthermore, according to Anderson et al. [27], important reasons for the filtering out of markers are lower variability, large amounts of missing data and genotyping errors. Thus, all the 7244 SNP and 4443 DArT markers retained passed the filtering criteria and were deemed appropriate for assessing the genetic diversity and structure of the studied soybean accessions.

From the SNP marker distribution, chromosome 12 had the least number of polymorphic SNP and DArT markers. Fewer polymorphic markers on chromosomes have been reported to be an indication of lower genetic diversity [24,28]; thus, chromosome 12 was identified as a less diverse chromosome in the soybean accessions. The PIC was used to estimate how informative the SNP and DArT markers were. Approximately 58% and 68% of the SNP and DArT markers, respectively, had PIC values within the range of 0.30–0.50, and they were therefore considered informative, revealing a higher genetic diversity compared to that in a previous report [9]. These results showed the applicability of SNP and DArT markers for better characterization of collections and for other genomic studies in soybean.

The cluster and PC analyses identified two groups that were similar to the model-based population structure analysis for the 281 accessions. The highest level of structure observed (k = 2) was similar to the structure reported by [9]. This similarity may be as a result of the predominance of the USA and Chinese accessions in the current study and the previous report. However, a population structure of five sub-groups has been reported [10] in soybean germplasms that comprise mostly South Korean accessions. Thus, it can be inferred that there is a low diversity between the Chinese and USA accessions. Karikari et al. [11] also reported a population structure of soybean comprising USA and Asian accessions with a few accessions from Africa, with four sub-groups. The mean *Fst* values of 0.23 (SNP Q1), 0.40 (SNP Q2) and 0.32 (DArT Q1) showed the existence of genetic diversity within these sub-populations, and parental materials can be selected to develop variable populations for breeding programs. Wright [29], suggested that an *Fst* value of 0.25 is an indication of good differentiation between population groups, from 0.15 to 0.25 is an indication of moderate differentiation and the differentiation is negligible if the *Fst* value is 0.05 or less. The similar grouping of some accessions from different countries indicated their close relatedness through common ancestry but may also be attributed to improper documentation during germplasm exchange; thus, the same accession may be recorded with different origins [30]. Hybridization between accessions in sub-populations 1 and 2 could offer the needed variation for enhancing genetic gain through active selection. 

The AMOVA revealed that 97% and 98% of all the genetic variances using SNP and DArT markers were within the population, and only 3% were attributed to population structure (variance among populations). Previous studies on soybeans [9,10] and other crops, such as *Camelina sativa* [31], wheat [28], Bangladesh rice [32] and rice [24] attributed a larger proportion of genetic variation to individual differences rather than the variance among populations.

The PhiPT value from the AMOVA showed that there were significant differences between the groups (PhiPT = 0.03 and 0.03, *p* = 0.001, for SNP and DArT markers, respectively). These significant differences may be as a result of the low genetic exchange and gene flow which was evident from the low value of the number of migrants. A low value for the number of migrants has been described as evidence of significant differences among populations [28,31]. Furthermore, the differentiation (structures) may be as a result of selection, genetic drift [33,34] and gene exchange [28,31], which will have impacts on the use of these populations in plant breeding programs [28]. In genome-wide association studies, the identification of quantitative trait loci and genomic selections, for assessing the patterns of genetic structure, is vital. The information obtained from such an assessment increases the confidence in the results of the associations detected.

The levels of diversity among population groups have been estimated using indices such as Shannon’s information index, number of private alleles, number of different alleles, number of effective alleles, diversity and unbiased diversity [24]. A higher value for these indices indicates a higher level of genetic diversity [28,31]. Based on these indices, the Chinese and USA accessions had higher diversity, while the accessions from Chile had the lowest diversity. In our study, these indices also revealed the existence of genetic diversity among the populations. The level of diversity identified may be attributed to the low gene-flow index [35] and represents valuable resources for future soybean improvement programs. The information revealed in this study could be useful for soybean breeding efforts in Africa. 

## 4. Materials and Methods

### 4.1. Plant Materials

The plant materials used in this study consisted of 281 soybean accessions from different countries, including improved cultivars (Appendix A). The accessions were collected from IITA. Seeds of each of the accessions were packed and sent to SEQART AFRICA, Nairobi, Kenya, for GBS using DArT [36,37]. SEQART AFRICA is a non-profit organization formed from the integration of DArT Pty Ltd., Canberra, Australia and the Biosciences eastern and central Africa (BecA) ILRI under the Integrated Genotyping Service and Support (IGSS) platform.

### 4.2. DArT-Based Genotyping by Sequencing

Genotyping of the soybean accessions was performed using the DArT-based GBS platform of SEQART AFRICA. Briefly, genomic DNA was extracted from young, fresh soybean leaves using the NucleoMag Plant genomic DNA extraction kit (Takara Bio, Shiga, Japan). The quality and quantity of DNA in each sample were checked via 0.8% *w*/*v* agarose gel and a NanoDrop spectrophotometer, respectively. Genomic DNA was digested using the DArT-seq complexity-reduction method and ligated to the barcoded adapters, after which PCR amplification of adapter-ligated fragments was performed. The genotypic analysis and SNP calling were carried out using the DArT-based GBS platform. The procedures for DNA extraction, DNA quality assessment, cleaning, complexity reduction, cloning and library construction have been previously described [37]. Samples were sequenced using a HiSeq 2500 system [36] and the final SNPs were called using the DArTsoft analytical pipeline (http://www.diversityarrays.com/darttechnologypackage-dartSoft, accessed on 11 October 2021).

### 4.3. Statistical Analysis

#### 4.3.1. SNP Filtering and Genome Characterization

The SNP characteristics (missing rate, PIC and frequency of heterozygosity) were generated by the DArT GBS platform. The filtering of the SNPs was performed in two stages; the first stage filtering removed all SNPs that were unmapped to any of the 20 soybean chromosomes, while the second stage filtering removed the redundant markers. The redundant markers were markers with missing rates of more than 10%, MAF below 5% and PIC < 10%. The genome summary plugin in the TASSEL v.5.2.37 software [38] was used to generate the allele frequency and MAF of the markers.

#### 4.3.2. Cluster and Population Structure Analyses

Cluster and principal component analyses of the 281 tropical soybean accessions were performed in TASSEL v.5.2.37 software [38] using the neighbour-joining algorithm. The grouping of accessions into clusters was achieved using the unweighted pair-group method with arithmetic means (UPGMA) and displayed via the Archaeopteryx Tree plugin.

The structure of the populations was assessed using a model-based (Bayesian) analysis in STRUCTURE v2.3.4 [39]. The k-values generated were from four independent analyses of the fixed number of population sub-groups (1–10). The analysis was based on 100,000 Markov chain Monte Carlo (MCMC) iterations and an initial burn-in period of 10,000, with no previous information on the source of accession. The best-fit k-value for the population was calculated using Structure Harvester [40]. In addition, Structure Harvester was used to generate the CLUMPP files for each k-value based on the log probability of the data [LnP(D)] and derived statistics (ΔK). The number of the sub-populations was determined when the ΔK value reached a maximum [41]. The individual Q-values for the accessions were obtained from the generated CLUMPP file, and accessions with Q-values > 0.5 are in the same group.

#### 4.3.3. Analysis of Molecular Variance

Analysis of molecular variance (AMOVA) was conducted using the results of population structures to determine the variation within and among the populations using GenAlEx 6.5 [42]. Genetic differentiation between populations was determined using the phi-statistics (PhiPT) value. The number of migrants (Nm value) was also estimated. The probability value used to test the significance of the variance was estimated using 1000 standard permutations. 

#### 4.3.4. Analysis of Allelic Pattern

The allelic pattern analysis across the population was performed using GenAlEx 6.5 [42]. The Shannon’s information index (I), diversity (h), unbiased diversity (uh), number of private alleles (Pa), number of different alleles (Na) and number of effective alleles (Ne) were used to assess the genetic diversity within and among the populations [40].

## 5. Conclusions

The use of different marker types in evaluating the genetic diversity of crops provides opportunities for a comparison of the marker types and gives a more reliable estimate of the diversity present. In the present study, the estimated genetic diversities were comparable between SNP and DArT markers. The few inconsistencies observed may be a result of the characteristics of the markers. The SNP and DArT markers provided similar evidence regarding the genetic diversity and population structure of the accessions. The DArT GBS platform produced high-density markers that were distributed across the 20 chromosomes of the soybean genome. Both markers revealed two population structures for the 281 soybean accessions. Greater genetic diversity existed within the population than among populations; thus, these populations could be useful in breeding programs aimed at recombining favourable alleles within adapted gene pools for the improvement of grain yield, stress resistance and seed quality traits. Additionally, the introgression of novel genes from diverse sources using advanced genomic techniques such as CRISPR/Cas9 technology, RNA interference (RNAi) and overexpression could be explored to develop more diverse populations. Our study provides an understanding of the genetic structure and diversity of the IITA soybean accessions, which will allow for the efficient utilization of these accessions in soybean improvement programs, especially in Africa.

Our next objectives are to identify promising genotypes and to conduct genome-wide association studies for seed quality traits, yield and biotic and abiotic stresses.

## Figures and Tables

**Figure 1 plants-11-00068-f001:**
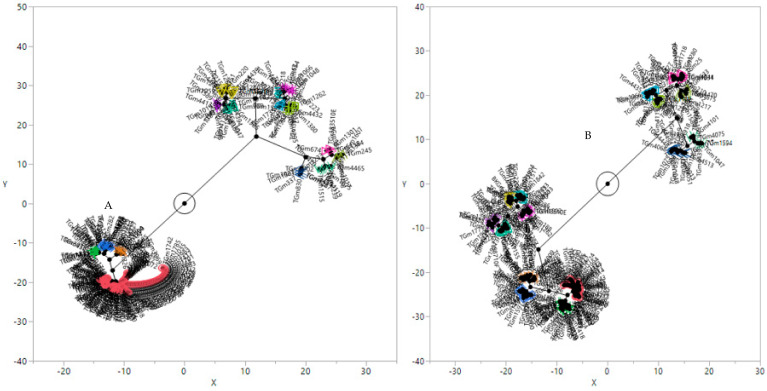
Cluster analysis of the 281 soybean genotypes based on unweighted pair-group method with arithmetic means: (**A**) SNP markers and (**B**) DArT markers. The different colors indicate the different clusters identified.

**Figure 2 plants-11-00068-f002:**
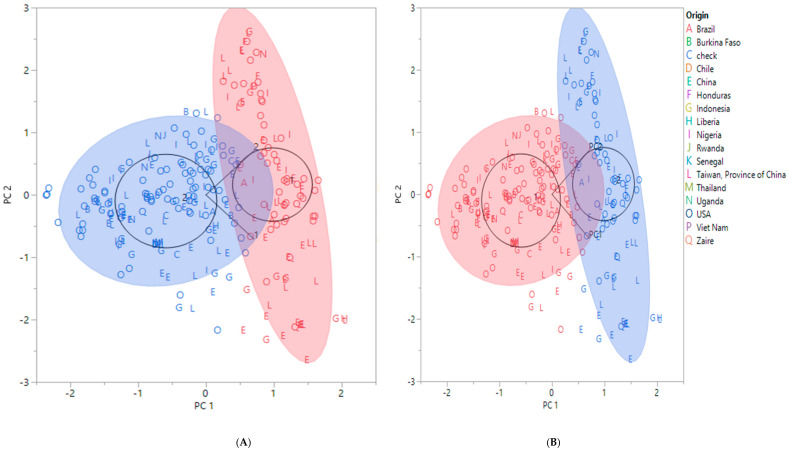
Principal component analysis biplot showing the first two components: (**A**) SNP markers and (**B**) DArT markers. The blue- and red-filled circles indicate the two groups identified. The black circles also represent the two groups and show the labels of the groups (1 and 2). The black line indicates the sub-group.

**Figure 3 plants-11-00068-f003:**
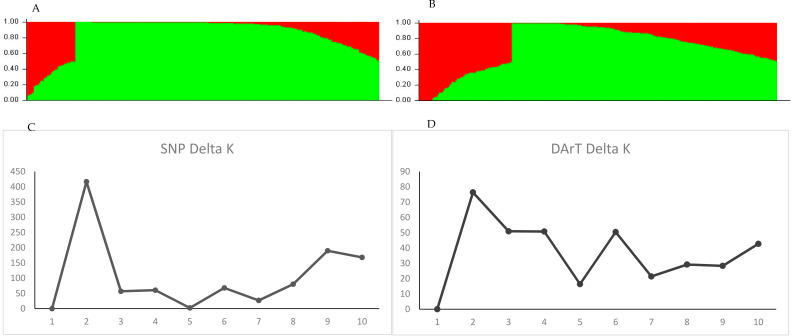
Population structure of the soybean accessions: (**A**) summary plot of estimates of Q for SNP markers; (**B**) rate of change of the likelihood (∆K) for SNP markers; (**C**) summary plot of estimates of Q for DArT markers; (**D**) rate of change of the likelihood (∆K) for DArT markers. The ∆K shows a clear peak at the true value of K.

**Table 1 plants-11-00068-t001:** Model-based Bayesian analysis of the SNP and DArT markers.

	Cluster Value	Mean Value of *Fst*	Expected Heterozygosity	Net Nucleotide Distance
SNP				
Q1	0.17	0.23	0.45	0.10
Q2	0.83	0.40	0.33	0.10
DArT				
Q1	0.32	0.32	0.27	0.09
Q2	0.63	0.19	0.30	0.09

*Fst* = Fixation index.

**Table 2 plants-11-00068-t002:** Analysis of molecular variance (AMOVA) for 281 soybean germplasms based on SNP markers.

Source	df	SS	MS	Estimated Variance	%
Among Pops	13	27,640.82	2126.227	40.45	3%
Within Pops	267	391,863.18	1467.65	1467.65	97%
Total	280	419,504.00		1508.10	100%

Stat	Value	P(rand ≥ data)			
PhiPT	0.027	0.001			
Nm (Haploid)	18.02				

Nm = number of migrants.

**Table 3 plants-11-00068-t003:** Analysis of molecular variance (AMOVA) for 281 soybean germplasms based on DArT markers.

Source	df	SS	MS	Estimated Variance	%
Among Pops	13	15,980.95	1229.30	22.21	2%
Within Pops	267	231,682.11	867.72	867.72	98%
Total	280	247,663.06		889.93	100%

Stat	Value	P(rand ≥ data)			
PhiPT	0.025	0.001			
Nm (Haploid)	19.54				

Nm = number of migrants.

## Data Availability

Not applicable.

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
