# Peer review of "Assessment of the Genetic Structure and Diversity of Soybean (Glycine max L.) Germplasm Using Diversity Array Technology and Single Nucleotide Polymorphism Markers"

_plants, 2021, doi:10.3390/plants11010068_

Round 1

Reviewer 1 Report

The manuscript entitled “Assessment of the Genetic Structure and Diversity of Tropical Soybean (Glycine max L.) Germplasm using Diversity Array Technology-based Single Nucleotide Polymorphism Markers” by Shaibu et al. described the genetic diversity and structure of a soybean germplasm collection by using and comparing DArT and SNP molecular markers.

Although the manuscript can be of large interest, I have in mind some significant concerns that the Authors must take into account in order to improve the manuscript making it suitable for publication.

First of all, I retain that the Authors have to move the figures of clustering in the manuscript while the current Figures 1 and 2 can be moved into the supplementary figures. This request because I retain that the cluster analysis must be well presented and discussed (e.g. are the mm able to distinguish the clusters for their geographic origin or other important classification?). The captions of the “new” figures on clustering included in the new version of the manuscript have to be improved.

In the cited literature, I did not find some important papers about genetic diversity and structure on soybean for which it is important to compare the results from this manuscript with those by Jeong et al. Theor Appl Genet. 2019; 132: 1179-1193 and Liu et al. Front Plant Sci. 2017; 8:2014.

In this respect, the Authors have to clarify the novelty of their results comparing other similar papers already published in the focus of the genetic diversity in this species. It will be interesting to deepen the aspect related to this very similar genetic structure of soybean. Did other authors find genetic structures more complex than K=2?

Finally, I found the English style of the manuscript really improvable and I warmly suggest the Authors to take care for this aspect in the revised version of their manuscript.

Author Response

Dear Reviewer,

Thank you for handling of our manuscript. We have incorporated all the suggestions and addressed all comments.

Thank you

Reviewer 2 Report

The present paper deals with development of SNP and DArT markers in soybean germplasms collected from various countries, USA, Asia and Africa. It might be an original work by the authors. However, the reviewer considers that there are many problematic points in the manuscript as shown below. The present manuscript might be hard to be published in the journal. It should be revised and submitted again.

  • Title: It is strange for the reviewer that the material accessions collected and maintained in IITA are regarded as “tropical germplasm.” Their origins are not in tropical areas, but mainly other countries. They should not be called “tropical germplasms.”
  • DArT markers and SNP markers are separately described in the text. Therefore, the title should be changed to “… Diversity Array Technology and Single Nucleotide polymorphism Markers.”
  • Abstract: The sentences are not concrete.
  • Line 21: What is the meaning of “reasonably distributed”? It is not scientific.
  • Line 26: What is the meaning of “somewhat different”? Where the differences are found?
  • Line 27: What is the meaning of “a certain amount of diversity”?
  • What is the meaning of “lack of structure and ability to distinguish accessions”?
  • Results: As a whole, only important numerals in the figures and tables are described in the text. However, interpretations of these numerals are not explained. It is important to lead the readers for understanding the numerals in the tables and figures; what do they mean?
  • In the Figures 1, 2 and 4, the meanings of the indices on the horizontal and vertical axes are not written in the figures.
  • Explain the results. How do you interpret the distribution of the markers on the chromosomes in Figure 1 ? And how do you explain the distribution of the markers in Figure 2 ?
  • Line 79: What is the meaning of “reasonably distributed”?
  • Line 94: Describe what is clarified in the Figure S1, S2 and . Explanation on PC analysis in Figure 3 is necessary.
  • Line 95: Figure S1 and S2 should be moved into the text. In Figure S1, the letters in the figures are hard to recognize. They should be deleted. What are the colors? More detailed explanation is necessary. What are numerals on the x and Y axes?
  • Line 111: What are individuals, accessions? What are the characteristics of the Q1 and Q2 ?
  • Line 119: It is hard to understand the meaning of “population structure.”
  • In Table 2, Upper part and lower part should be separated.
  • Discussion: Line 183: What is the meaning of “may be considered informative”?
  • Line 198-202: What does it mean that 97 and 98% of the genetic variances were within the population and 3 and 2% were attributed to population structure? The population means 281 germplasms, right?
  • Line 219: What does “somewhat different” mean? As the conclusion, what is the genetic structure of the population of the present germplasms investigated?

Author Response

(The authors gave the same response as above.)

Round 2

Reviewer 1 Report

The revised version of manuscript entitled “Assessment of the Genetic Structure and Diversity of Tropical Soybean (Glycine max L.) Germplasm using Diversity Array Technology-based Single Nucleotide Polymorphism Markers” by Shaibu et al. has been improved following the suggestion by the reviewer except two important corcerns: the Authors did not report a detailed comparisons between their work and those already published (e.g. is this manuscript able to distinguish the accessions for their origin? Are previously reported genetic structure more complex then K=2 reported in their work?). The English style of the manuscript was not improved at all. 

Author Response

Thank you very much for your valuable suggestions/comments to assist us to improve the scientific quality of our manuscript. We have compared the population structure of our work with previous studies (see Line 220-226). We have further improved the English style to the best of our ability. In addition, two colleagues of us who are native English speakers have read and edited where ever necessary.

Reviewer 2 Report

The revised manuscript was very much improved and became easy to read. However, there are still some points to be clarified. The reviewer expects the authors to revise the manuscript to be understood by readers more accurately and deeply.

1) The 281 soybean germplasms were grouped into fourteen populations depending on their original areas. However, this grouping was not shown in Materials and Methods. Therefore, it is very difficult to understand the AMOVA. List of the materials in Table S1 should be changed according to the grouping of the germplasms into fourteen populations (Population 1 to 14).

2) In Figure 1 and Figure 2, the 281 germplasms were divided into two groups and again divided into several subgroups. Describe which accessions are belonging to which subgroups. Describe what does this fact means, and how are these data interpreted. What are the characteristics of the subgroups? Without these kinds of interpretation of the data, there are no meaning of the analyses.

3) Line 183: In the explanation of Figure 1, ‘… 282 soybean genotypes …’ is written. It must be ‘281 genotypes’.

4) Also in Figure 3, it is described that most of the African germplasms were in Q2 and others were in Q1. What does this fact mean? 

5) Line 131: What is ‘individuals’? Does it mean individual germplasm? -> The germplasms in Q2 were sub-divided into ….

6) Line 223: including those from Africa …

7) Line 254: ‘181 accessions’ should be ‘281 accessions’.

8) Line 292: … germplasms …

9) Line 329: … germplasms …

Author Response

The revised manuscript was very much improved and became easy to read. However, there are still some points to be clarified. The reviewer expects the authors to revise the manuscript to be understood by readers more accurately and deeply.

Response: We appreciate your effort to assist us to improve the readability and clarity of our manuscript. We have further revised the manuscript with much emphasis on the interpretation and discussion of the population structure results. In addition, two colleagues of us who are native English speakers have read and edited to improve the clarity of reading.

1) The 281 soybean germplasms were grouped into fourteen populations depending on their original areas. However, this grouping was not shown in Materials and Methods. Therefore, it is very difficult to understand the AMOVA. List of the materials in Table S1 should be changed according to the grouping of the germplasms into fourteen populations (Population 1 to 14).

Response: We have added the information about the origin to Table S1 in the revised manuscript.

2) In Figure 1 and Figure 2, the 281 germplasms were divided into two groups and again divided into several subgroups. Describe which accessions are belonging to which subgroups. Describe what does this fact means, and how are these data interpreted. What are the characteristics of the subgroups? Without these kinds of interpretation of the data, there are no meaning of the analyses.

Response: Figures 1 and 2 have been interpreted and discussed further in the revised manuscript.

3) Line 183: In the explanation of Figure 1, ‘… 282 soybean genotypes …’ is written. It must be ‘281 genotypes’.

Response: We have replaced 282 in Line 183 with 281 in the revised manuscript (see Line 160). Thank you.

4) Also in Figure 3, it is described that most of the African germplasms were in Q2 and others were in Q1. What does this fact mean? 

Response: We have modified it to bring more clarity to Figure 3.

5) Line 131: What is ‘individuals’? Does it mean individual germplasm? -> The germplasms in Q2 were sub-divided into ….

Response: We have revised to bring more clarity to the sub-groupings of the accessions in the revised manuscript (See Line 111-112). Thank you.

6) Line 223: including those from Africa …

Response: The correction has been incorporated in the revised manuscript (See Line 195). Thank you.

7) Line 254: ‘181 accessions’ should be ‘281 accessions’.

Response: We have replaced 181 in Line 254 with 281 in the revised manuscript (see Line 220). Thank you.

8) Line 292: … germplasms …

Response: We have replaced germplasms with accessions in the revised manuscript.

9) Line 329: … germplasms …

Response: We have replaced germplasms with accessions in the revised manuscript.